# Machine and deep learning algorithms for sentiment analysis during COVID-19: A vision to create fake news resistant society

**Muhammad Tayyab Zamir**[1], **Fida Ullah**[1], **Rasikh Tariq**[2]*, **Waqas Haider Bangyal**[3], **Muhammad Arif**[1], **Alexander Gelbukh**[1]*

**1** Centro de Investigación en Computación (CIC), Instituto Politécnico Nacional, Ciudad de México, México, **2** Tecnologico de Monterrey, Institute for the Future of Education, Monterrey, N.L., México, **3** Department of Computer Science, Kohsar University Murree, Pakistan

* gelbukh@gelbukh.com, gelbukh@cic.ipn.mx (AG); rasikhtariq@tec.mx (RT)

**Data Availability Statement:** All code files are available from the https://github.com/

## Abstract

Informal education via social media plays a crucial role in modern learning, offering self-directed and community-driven opportunities to gain knowledge, skills, and attitudes beyond traditional educational settings. These platforms provide access to a broad range of learning materials, such as tutorials, blogs, forums, and interactive content, making education more accessible and tailored to individual interests and needs. However, challenges like information overload and the spread of misinformation highlight the importance of digital literacy in ensuring users can critically evaluate the credibility of information. Consequently, the significance of sentiment analysis has grown in contemporary times due to the widespread utilization of social media platforms as a means for individuals to articulate their viewpoints. Twitter (now X) is well recognized as a prominent social media platform that is predominantly utilized for microblogging. Individuals commonly engage in expressing their viewpoints regarding contemporary events, hence presenting a significant difficulty for scholars to categorize the sentiment associated with such expressions effectively. This research study introduces a highly effective technique for detecting misinformation related to the COVID-19 pandemic. The spread of fake news during the COVID-19 pandemic has created significant challenges for public health and safety because misinformation about the virus, its transmission, and treatments has led to confusion and distrust among the public. This research study introduce highly effective techniques for detecting misinformation related to the COVID-19 pandemic. The **methodology** of this work includes gathering a dataset comprising fabricated news articles sourced from a corpus and subjected to the natural language processing (NLP) cycle. After applying some filters, a total of five machine learning classifiers and three deep learning classifiers were employed to forecast the sentiment of news articles, distinguishing between those that are authentic and those that are fabricated. This research employs machine learning classifiers, namely Support Vector Machine, Logistic Regression, K-Nearest Neighbors, Decision Trees, and Random Forest, to analyze and compare the obtained results. This research employs Convolutional Neural Networks, Long Short-Term Memory (LSTM), and Gated Recurrent Unit (GRU) as deep learning classifiers, and afterwards compares the obtained results. The **results** indicate that the BiGRU deep

tayyabawan786/covid19-project/tree/main
database.

**Funding:** The authors would like to thank
Tecnológico de Monterrey for the financial support
provided through the 'Challenge-Based Research
Funding Program 2023', Project ID #IJXT070-
23EG99001, entitled 'Complex Thinking Education
for All (CTE4A): A Digital Hub and School for
Lifelong Learners.' The authors acknowledge the
financial support of Writing Lab, Institute for the
Future of Education, Tecnologico de Monterrey,
Mexico, in the production of this work.

**Competing interests:** The authors have declared
that no competing interests exist.

learning classifier demonstrates high accuracy and efficiency, with the following indicators: accuracy of 0.91, precision of 0.90, recall of 0.93, and F1-score of 0.92. For the same algorithm, the true negatives, and true positives came out to be 555 and 580, respectively, whereas, the false negatives and false positives came out to be 81, and 68, respectively. In **conclusion**, this research highlights the effectiveness of the BiGRU deep learning classifier in detecting misinformation related to COVID-19, emphasizing its significance for fostering media literacy and resilience against fake news in contemporary society. The implications of this research are significant for higher education and lifelong learners as it highlights the potential for using advanced machine learning to help educators and institutions in the process of combating the spread of misinformation and promoting critical thinking skills among students. By applying these methods to analyze and classify news articles, educators can develop more effective tools and curricula for teaching media literacy and information validation, equipping students with the skills needed to discern between authentic and fabricated information in the context of the COVID-19 pandemic and beyond. The implications of this research extrapolate to the creation of a society that is resistant to the spread of fake news through social media platforms.

## 1. Introduction

In the contemporary era, our world revolves around the internet, a technology that swiftly conveys information across the globe. The internet is the fastest channel for information dissemination, making it imperative to examine its advantages and disadvantages. At the core of this technological landscape are social networks, which have revolutionized information access owing to their online presence [1, 2]. Over recent years, access to information has accelerated, driven by advances in internet technology and the production of diverse information sources through platforms such as Facebook, Instagram, Twitter (now X), WhatsApp, WeChat, Google, and various microblogging sites. These platforms enable global connections and data sharing at a minimal cost which is a considerable advantage of social networks [3]. However, recognizing the accuracy of information within them often proves challenging, as it may contain misinformation or rumors.

On the other hand, informal education through social media networks constitutes a significant aspect of contemporary learning, encompassing the acquisition of knowledge, skills, and attitudes outside traditional educational frameworks. This mode of education is characterized by its non-linear, self-directed, and often community-driven nature, enabling individuals to explore diverse interests at their own pace and on their own terms. Social media platforms facilitate a wide array of informal learning opportunities, from tutorials and expert blogs to discussion forums and interactive content, thereby democratizing access to information and expertise across global communities. Unlike formal education, which is structured and certification-oriented, informal learning via social media is fluid, with the learner navigating through content based on curiosity, personal or professional interest, and immediate relevance. This phenomenon significantly contributes to lifelong learning, fostering a culture of continuous personal and professional development. However, the informal education received through social media also presents challenges, including information overload, the spread of misinformation, and the need for critical thinking to discern the quality and credibility of information sources [4]. Similarly, Jin et al. [5] concluded that digital literacy significantly reduces the

likelihood of fraud among residents of six East Chinese provinces, particularly benefiting rural, middle-aged, elderly, females, and financially educated individuals. Therefore, while social media networks serve as potent vehicles for informal education, they underscore the necessity for digital literacy skills to navigate this landscape effectively.

It is essential to verify the legitimacy of available data to prevent the spread of false information, as unfounded rumors can significantly affect individuals' personal and social lives. This statement finds support in the research conducted by Islam et al. [6] which underscored the widespread dissemination of COVID-19-related misinformation, encompassing rumors, stigma, and conspiracy theories, across various online platforms. The study also underscored the potentially severe implications for public health, emphasizing the necessity for real-time monitoring and active engagement with communities and government stakeholders to counter the proliferation of false information. On the other hand, Loomba et al. [7] revealed that achieving high COVID-19 vaccine uptake is crucial in combating the pandemic which may be hindered by online misinformation, as exposure to such information was found to reduce intent to vaccinate in both the UK and the USA, particularly among those who were initially willing to take the vaccine, and certain demographic groups are more affected by this misinformation. In this context, sentiment analysis has critical significance in today's world, where the spread of misinformation, particularly in the context of health and public information, can significantly affect individuals' lives and influence public health outcomes, making it essential for real-time monitoring and informed engagement with communities and stakeholders to counter false information and promote informed decision-making.

## 1.1. Background

Sentiment analysis has gained significant importance in the current era due to the global utilization of various social media platforms as a means for individuals to express their opinions. Individuals engage in self-expression by posting comments on various social media platforms such as Instagram, Facebook, Twitter, and others [8]. The individuals discuss many subjects, such as films, commodities, prevailing fashion, political affairs, and technological advancements. The individual's viewpoint can be categorized as good, negative, neutral, or spam. Multiple polarity scales can be utilized to predict scores [9]. There should be a process to further expound upon these comments, which is sentiment analysis.

Sentiment analysis [10], a component of Natural Language Processing, is commonly referred to as opinion mining due to its focus on examining users' emotions, feelings, and opinions. Sentiment analysis serves a broader range of applications beyond opinion prediction, encompassing the identification of fraudulent tweets [11]. The user's text does not contain any information. The categorization methodologies employed in this study consist of three levels: feature, document, and sentence [12].

The application of sentiment analysis necessitates the utilization of not one but two separate methodological approaches. The first dictionary or lexicon is based on traditional methods, whereas the second dictionary or lexicon is based on techniques used in machine learning [13–17]. The ability of machine learning techniques to process and successfully evaluate large datasets in an automated manner has led to their increased prevalence in today's technological landscape. Several different algorithms based on machine learning are utilized to perform sentiment analysis, and these approaches have shown promising results. Techniques such as the Naive Bayes classification, Bootstrap aggregation, Linear Discriminant Analysis, Decision Trees, and Support Vector Machines are some examples of this type of methodology [18]. Methodologies for machine learning (ML) can be broken down into one of three categories: unsupervised, semi-supervised, or supervised [19]. When it comes to sentiment analysis,

supervised learning algorithms are extremely popular and have a high level of regard. The implementation requires the extraction of sentences and features at several different levels, including feature level and sentence level. The feature set consists of a bag of words as well as tags for the various parts of speech, unigrams, bigrams, and n-grams.

The limitations that are imposed on the processing capabilities of machine learning have led to the development of a more sophisticated strategy known as deep learning (DL). In 2006, G.E. Hinton was the first person to suggest the idea of "Deep Learning." He did so by introducing a representation of a Machine Learning job that specifically pertains to Deep Neural Networks [13]. When he did this, he also created the concept of "Deep Learning." The functional model of the human brain is used as the basis for the construction of the Artificial Neural Network (ANN) [20]. This model describes how several neurons work together to facilitate the processing of information. The capacity of Deep Learning models to provide training for both supervised and unsupervised categories is one reason why these models are used. Word representation, sentence categorization, and text synthesis are three examples of the natural language processing tasks that have been shown to benefit significantly from the utilization of neural networks. Deep learning has played an important role in the expansion of computational capabilities as well as the development of the field of software engineering [16]. Because of its widened computing capabilities, the system is able to successfully evaluate and manage huge amounts of data utilizing several neural networks, such as the Deep Forward Neural Network, which has led to a significant improvement in accuracy [21, 22]. Convolutional Neural Networks (CNN) [23], Recurrent Neural Networks (RNN), Deep Belief Networks (DBN), Long Short-Term Memory (LSTM), and Bidirectional LSTM (BiLSTM [24]), amongst other methods, are all utilized in Deep Learning for the purpose of doing sentiment analysis. F-measure, recall, and precision are the objective measures used by machine learning and deep learning algorithms to analyze the effectiveness of a classifier. These three criteria help select the best appropriate model to address a specific issue and determine how well it addresses it.

## 2. Literature review and related works

### 2.1. Machine learning and deep learning algorithms for sentiment analysis

There exists quite a lot of literature in the current state of the art which has focused on the application of machine learning or deep learning methods for sentiment analysis on Twitter [25] examined sentiment in large-scale social data. Social media platforms are becoming more popular for expressing opinions on many topics. Integrated CNN, LSTM, and DNN (Deep Neural Network) are used in this article to provide a new sentiment analysis strategy. ICNN-LSTM-DNN is used for social data sentiment analysis and opinion mining during the 2019 Indian general election. This study tests the algorithm using twitter (now X) data. The above strategy was 89% accurate. On the other side, Gopi et al. [18] studied the Twitter (now X) sentiment analysis. Data polarity is determined in sentiment analysis. Polarity definitions are often asked. Positive, negative, or neutral data is used in polarity opinion mining. In this study, researchers predicted sentiment using a multi-polarity scale. The scale ranged from -5 to +5. Another prominent research by Ali et al. [26] identified fraudulent tweets. Twitter (now X) is the most popular microblogging network, allowing people to share their opinions on a variety of topics with substantial weight and effect. Numerous bogus accounts influence the topic's conversation. These accounts must be distinguished from legitimate and authorized ones. The classifier flags fake accounts using a Custom Rule-Based algorithm. Other machine learning methods including Multi-Layer Perceptron (MLP), Decision Tree, and Random Forest are used and contrasted. The researchers used Twitter (now X) API (application programming interface) user data. A data frame was created from up to 200 tweets per person. The approach

outperformed MLP, Decision Tree, and Random Forest with 97% accuracy on the same dataset. The current body of research is replete with numerous studies on sentiment analysis.

In the landscape of COVID-19 research, multiple articles delve into different facets, including prediction, diagnosis, and sentiment analysis. Indumathi et al. [27] proposed a machine learning algorithms to predict COVID-19 affected zones in Virudhunagar district, achieving an impressive accuracy of 98.06%. Ayalew et al. [28] introduce a unique detection and classification approach (DCCNet) for COVID-19 using chest X-ray images, exhibiting a remarkable 99.9% training accuracy and 98.5% test accuracy. Prasad et al. [29] suggest a cloud-based image analysis approach (CIA-CVD) for COVID-19 vaccine distribution, integrating deep learning, machine learning, digital image processing, and cloud solutions. Misra et al. [30] present a Parallel Ensemble Transfer Learning based Framework for COVID (PETLFC) utilizing chest X-ray images, demonstrating superior performance compared to sequential ensemble approaches. Ayalew et al. [31] employ a Convolutional Neural Network (CNN) for COVID-19 detection from chest X-ray images, achieving high training and test accuracy. Salau et al. [32] roposes a Support Vector Machine (SVM) method with a discrete wavelet transform (DWT) algorithm, attaining a detection rate of 98.2%. Furthermore, Demilie and Salau [33] address the pervasive issue of hate speech detection on social media platforms, emphasizing the need for advanced research and optimal approaches in this challenging domain. Gupta et al. [34]contribute to the sentiment analysis domain, mining the sentiments of Indian citizens regarding the nationwide lockdown during the COVID-19 outbreak, with a notable accuracy of 84.4%. Jayasurya et al. [35] conduct sentiment analysis on the topic of COVID-19 vaccination, employing 14 different machine learning classifiers and revealing insightful temporal and spatial analyses of textual data. Collectively, these articles underscore the interdisciplinary efforts required to comprehensively combat and understand the multifaceted impacts of COVID-19, utilizing advanced technologies and methodologies across various domains, from healthcare to social analysis. A comprehensive summary of these studies is provided in Table 1.

## 2.2. Recent advances with Large Language Models

Large Language Models (LLMs) and transformers, including BERT, RoBERTa, LLMA 2, and GPT [5, 49], have emerged as essential tools in the domain of fake news detection, demonstrating substantial contributions to improving detection accuracy. Szczepański et al. [50] provided an in-depth study of BERT's performance, introducing an innovative approach to enhance explainability in fake news detection. Their model utilized Local Interpretable Model-Agnostic Explanations (LIME) and Anchors, enabling users to understand the rationale behind BERT's classification decisions. This advancement emphasized the need for transparency in AI-driven systems, highlighting that high accuracy alone is insufficient in critical real-life applications, such as disinformation detection. Through experimentation with fake news datasets, including tweets and headlines, they demonstrated that these explainability techniques could be seamlessly integrated into BERT models without affecting performance. Their findings showed that explainability enhances user trust, particularly in high-stakes environments, where understanding decision-making processes is as important as the outcome itself.

RoBERTa, an optimized version of BERT, further advanced the field by refining pretraining on larger datasets and for extended durations. Pavlov et al. [51] explored RoBERTa's effectiveness in detecting fake news specifically related to COVID-19. By fine-tuning RoBERTa on a dataset comprising real and fake COVID-19-related tweets, they achieved notable improvements in accuracy, recall, and F1 score when compared to standard BERT models. Their research revealed that RoBERTa's ability to capture subtle linguistic nuances in

**Table 1. Current state-of-the-art around sentiment analysis.**

| Reference | Description | Classifier algorithm | Results |
|---|---|---|---|
| [21] | In this research, scientists used Deep Learning to enhance sentiment analysis of English and Korean tweets gathered via the Twitter API. | The following algorithms are employed on this work:<br>Deep Neural Network<br>Multilayer Perceptron | Deep Neural Networks (DNN) has 75.03% accuracy on testing dataset, while the MLP had 52.60%. |
| [36] | Researchers investigates Twitter data-driven emotion recognition with an emphasis on real-time classification to improve user experience | The following algorithms are used: stochastic gradient decent multinomial, naive bayes, Bernoulli naive bayes, Nearest Centroid, Ridge classifier, and Random Forest. | The accuracy attained by the state-of-the-art methods is 65.57%; this article increases it by 5.83%, yielding a new accuracy of 71.40%. |
| [37] | Researchers analyzed Twitter sentiment. The study classifies recent tweets as positive or unfavorable. It is well known that sentiment analysis uses numerous machine learning methods to determine polarity. | A sample of 100 recent tweets was tested. Sentiwordnet's Naive Bayes classifier determines tweet polarity. This method greatly enhances experimental precision. | 58.40% accuracy was achieved using Naive Bayes Classifier, which was one of the approaches employed along with Baseline and Support Vector Machine. |
| [38] | The study focuses on detecting spam tweets using N-gram features while also addressing the issue of harmful tweets. It involves implementing and comparing three machine learning classifiers in an experimental setting. | The machine learning classifiers utilized in this work are Support Vector Machine, Random Forest, and Logistic Regression. | The self-generated dataset was utilized for experimentation, and the outcomes were subsequently compared. It was observed that the N-gram approach yielded an accuracy rate of approximately 80%. |
| [39] | The study's objective was to use a Deep Learning methodology to analyze movie reviews and compare the results with other Deep Learning techniques. 50 000 movie reviews from the IMDB corpus, equally divided between favorable and unfavorable ratings, make up the dataset. | The present research investigation employs a specialized classifier known as the CNN-LSTM classifier, which combines CNN and LSTM models. | The hybrid classifier outperformed the other classifiers in terms of accuracy. The hybrid CNN and LSTM model demonstrated superior classification performance, with an accuracy of 89.2%. |
| [40] | Using data from Twitter, this study aims to understand the reactions of individuals to COVID-19 in eight different nations. Scientists discovered that different countries' reactions to the pandemic elicited different kinds of attention and emotions. | This study combined one traditional machine learning model with five deep learning models to analyze sentiment. Along with meta-learning approaches, the models that are used are CNN, BiGRU, FastText, DistilBERT, Naive Bayes, and Support Vector Machines | Among all models, the proposed model outperformed the others with the highest accuracy of 0.858. The accuracy of CNN was the next highest at 0.816, followed by that of BiGRU and FastText at 0.797 and 0.796, respectively. DistilBERT obtained 0.855 accuracy, whereas NBSVM registered 0.798. |
| [41] | This research study uses Twitter data to determine people's sentiment toward the news of COVID-19. They created the scenario using the R programming language. The data set was obtained via Twitter using hashtags such as COVID-19, disease, new incidence, coronavirus, etc. | In this work researchers used algorithms Support vector machine (SVM), Hybrid Heterogeneous Support vector machine(H-SVM) and RNN<br>Abbreviated as Recurrent Neural Network. | The MKH-SVM model achieved an accuracy of 96.3%, outperforming the Linear SVM model, which achieved 93.6%. The RNN model followed with a lower accuracy of 91%. |
| [42] | This research study was conducted on farmer demonstrations in India against questionable government measures. They conducted a study using Twitter data, as individuals indicated their support for or opposition to demonstrations on Twitter. | Their research study compared the outcomes of four different machine learning classifiers. They made use of DT, NB, SVM, and RF. | Naive Bayes achieved 72.9%, Decision Tree 79.78%, Random Forest 96.62%, and SVC 83.45%. The Random Forest model has the highest accuracy. |
| [43] | This research was focused on internet word of mouth since it greatly influences people's opinions and decisions. | In this work, they employed SVM, Naive Bayes, and KNN. | The suggested approach outperformed the existing ML classifiers with an accuracy of 89.2%. |
| [44] | This study was conducted using Twitter data for COVID-19 false news identification, and the data set was obtained from the corpus. | They employed eight ML classifiers and four DL classifiers in their work. | They employed eight ML classifiers and four DL classifiers in their work. |
| [45] | Based on the study conducted by [26], it has been determined that conventional natural language processing (NLP) techniques may not be sufficiently viable for use on large-scale datasets in order to uncover sentiments. Consequently, further advancements and investigations are necessary in this domain. | In this research study researchers suggested a model incorporated a series of consecutive convolutional layers, and its performance was evaluated against other machine learning approaches and deep learning techniques. This study examined four algorithms. | The classifier's results showed notable improvement when introducing a larger dataset, a noteworthy observation in this research study. The findings indicate that the suggested technique achieved an accuracy of 88.34%, surpassing the performance of all other algorithms. |

*(Continued)*

**Table 1.** (Continued)

| Reference | Description | Classifier algorithm | Results |
|---|---|---|---|
| [46] | The research introduces n-gram IDF (inverse document frequency) feature extraction-based machine learning. After feature extraction, machine learning was used to classify data by sentiment. This study used mobile app reviews, Jira comments, and Stack Overflow questions-and-answers. | For feature representation, n-gram IDF is used to extract software-engineering related, dataset-specific, positive, neutral, and negative n-gram expressions. For classifiers, an automated machine learning tool is used. | The proposed paradigm achieved the highest accuracy among all theories, with accuracy rates of 1317/1500, 293/341, and 884/926. These figures indicate strong performance across different datasets or scenarios. |
| [47] | The study shows that social media emotions have great potential for positive use. The researchers combined term frequency (TF) and term frequency-inverse document frequency (TF-IDF) feature extraction methods. | In this study the researchers used the Neural Networks, Ordinal regression and Binary-mapped regression models. | Study shows exceptional F-score of 85.16%, the classification model. Furthermore, sentimental correlation of 71.44% was shown. |
| [48] | This research was related to the stock market. The researchers developed a hybrid deep-learning model to analyze sentiment and forecast emotions among investors. | Long short-term Memory Neural Network And for Hybrid model CNN-LSTM were used in this study. | The experimental findings demonstrate that the proposed paradigm surpasses the model utilized in the baseline study in terms of its efficacy in predicting stock values. The proposed model demonstrates an accuracy rate of 84.7%. |
| This work | This research study introduces a highly effective technique for detecting misinformation related to the COVID-19 pandemic. The dataset is taken from publicly available repository Kaggle. | This work proposes the application of five machine learning classifiers SVM, LR, KNN, DR, and RF and three deep learning techniques CNN, LSTM and GRU. | The proposed model achieve the accuracy precision of 0.90, recall of 0.93, and F1-score of 0.92 and surpasses the other models |

misinformation made it particularly effective in distinguishing real from fake COVID-19 news. This study demonstrated that RoBERTa, due to its extended training and larger data exposure, not only enhances detection performance but also improves the robustness of the model in real-world, noisy social media environments. The comparative results with BERT underscore the advantages of extended pretraining in improving model generalization across diverse datasets.

In addition to detection, GPT models have been pivotal in simulating fake news generation, as demonstrated by Dhiman et al. [52]. Their work introduced GBERT, a hybrid deep learning framework that combines GPT's generative capabilities with BERT's deep contextual understanding. By fine-tuning both models on real-world benchmark datasets, the authors achieved superior accuracy (95.30%), precision (95.13%), and F1 score (96.23%). The generative aspect of GPT allowed the creation of high-quality, diverse fake news samples, which were then used to train the detection model, thus enhancing its ability to handle adversarial examples. This approach demonstrated the utility of combining generative and discriminative models to address the evolving challenge of misinformation, particularly in the era of LLMs. The statistical tests conducted in the study revealed that the hybrid model not only improved performance but also increased robustness against various types of fake news, establishing a promising direction for future research in the field.

Su et al. [49] identified another critical issue with LLM-generated text—bias in fake news detectors. Their research revealed that existing detection models tend to misclassify human-written fake news as genuine while disproportionately flagging LLM-generated content as fake. This bias was attributed to the distinct linguistic patterns produced by LLMs, which detectors could easily identify. To mitigate this, the authors employed adversarial training using LLM-paraphrased genuine news, significantly improving detection accuracy across both human- and machine-generated news. Their introduction of two novel datasets, GossipCop+ + and PolitiFact++, consisting of human-validated and LLM-generated news, provides a valuable resource for future studies. This work highlights the importance of addressing bias in fake news detection, particularly as LLMs become more adept at generating human-like content.

The recent advancements in fake news and fake review detection have further demonstrated the utility and limitations of transformer-based models. Qin and Zhang's study [53] explores the challenge of generalization in BERT fine-tuning for fake news detection, a common issue that arises due to overfitting on the training data. Their research identifies that BERT's fine-tuning often leads to reduced generalization capability, making it difficult to detect unseen fake news accurately. To address this, they propose a novel adversarial fine-tuning strategy, FGM-FRAT, which treats fine-tuning as a mini-max optimization problem. By applying adversarial training on the word vector space and incorporating feature regularization, the model enhances its robustness against unseen data. Extensive experiments on benchmark datasets, including FakeNewsNet and KaggleFakeNews, demonstrated that the proposed method improves generalization accuracy from 61% to 73%, significantly outperforming conventional fine-tuning methods. This shows that while BERT-based models have shown great promise, careful fine-tuning strategies are essential to extend their applicability to diverse datasets and new scenarios.

In contrast, Mohawesh et al. [54] tackle the issue of fake review detection by combining RoBERTa with an LSTM layer to build a more semantic- and linguistic-aware model. Their approach focuses on overcoming the limitations of traditional machine learning models, which primarily rely on linguistic features and often fail to capture deeper semantic meanings within text. The authors integrated RoBERTa's transformer capabilities with LSTM, the model can process elaborate temporal dependencies within the data. The authors further enhance the model's performance by incorporating Shapley Additive Explanations (SHAP) and attention mechanisms to ensure transparency in classification decisions. Their experiments on the OpSpam and Deception datasets demonstrated that this hybrid model outperformed state-of-the-art methods, achieving 96.03% and 93.15% accuracy, respectively. The use of SHAP and attention mechanisms also provided clearer insights into why specific reviews were classified as fake, adding an important layer of interpretability. This highlights the critical role that hybrid models and explainability techniques play in ensuring robust and transparent detection systems, especially in high-stakes contexts like consumer behavior.

In summary, the integration of Large Language Models like BERT, RoBERTa, and GPT into fake news detection has significantly advanced the field, offering both improved performance and new methodological insights. The works of Szczepański et al. [50], Pavlov et al. [51], Dhiman et al. [52], and Su et al. [49] demonstrate that these models enhance detection accuracy through deep contextual understanding and robust training, and on the other side, address critical issues such as explainability and bias.

## 2.3. Fake news resistant society through media literacy and artificial intelligence tools

Creating a society resistant to fake news necessitates a comprehensive approach to media literacy, emphasizing critical thinking, verification skills, and awareness of information sources. Media literacy education empowers individuals to critically evaluate content, discerning between credible information and misinformation. In this context, the integration of Artificial Intelligence (AI), particularly through sentiment analysis, can significantly enhance media literacy efforts. Sentiment analysis, a subset of AI, involves analyzing texts to determine the sentiment behind them, offering insights into the biases and emotional undertones that may indicate biased or manipulative content. By using AI tools that apply sentiment analysis, educators and learners can more effectively identify the emotional manipulation often associated with fake news. This not only facilitates a deeper understanding of how information is constructed to influence public opinion but also equips individuals with the analytical tools to

question and verify the authenticity of the content they encounter. Consequently, the use of AI through sentiment analysis in media literacy programs represents a promising avenue for fostering a society more resilient to the pervasive challenge of fake news, by enhancing the analytical capabilities essential for discerning the reliability and bias of information sources.

Scheibenzuber et al. [55] discuss the development and testing of an undergraduate online problem-based course designed to combat fake news illiteracy, especially during the Covid-19 pandemic. This course, developed in response to the surge in misinformation ("infodemic") and the shift to emergency online learning, aimed to enhance students' ability to critically assess the credibility of news. It incorporated insights from fake news research and the problem-based learning approach. The study, involving 102 students, found positive student feedback on online communication and feedback aspects of the course, though some suggested improvements for task descriptions. Significant improvements were observed in students' ability to assess the credibility of fake news, with academic achievements reflecting a good to very good standard. The conclusion underscores the potential of problem-based online courses as effective environments for improving fake news literacy, even in emergency learning contexts.

Mi and Apuke [56] explore the role of social media knowledge in combating fake news among Nigerian social media users. Their structural equation modeling analysis reveals that altruism and self-promotion significantly influence fake news sharing behavior. Moreover, individuals with higher social media knowledge are less likely to engage in fake news sharing. The study concludes that enhancing social media knowledge can effectively reduce the spread of misinformation, contributing to both theoretical understanding and practical approaches to curbing fake news dissemination.

Jang and Kim [57] investigate the third-person effect in the context of fake news, finding that individuals believe fake news has a greater impact on others than on themselves or their in-group. The study identifies partisan identity, social undesirability, and external political efficacy as predictors of this perception. Interestingly, those with a stronger third-person perception favor media literacy interventions over media regulation to combat fake news. This research highlights the varied responses to fake news and underscores the importance of media literacy as a preferred solution.

Li et al. [58] examine the public's engagement with fake news rebuttals on social media during the COVID-19 pandemic. Using the fake news rebuttals engagement index (FNREI) and time-varying parameter vector auto-regressive model (TVP-VAR), the study analyzes data from 19,603 Weibo posts. Findings reveal that public engagement with COVID-19 related fake news rebuttals varies over time and is influenced by major pandemic events. This research provides insights into how public engagement with fake news rebuttals evolves and offers suggestions for enhancing fake news governance effectiveness.

Beauvais [59] explores the reasons behind individuals' belief in fake news, particularly during the COVID-19 pandemic and US presidential elections. The study identifies media and social network ecosystems, cognitive and psychological factors, and sociological influences as key determinants. It suggests that high digital literacy can protect against believing in fake news and recommends measures for journalism, education, fake news detection, fact-checking, and social network regulation to counteract misinformation. This study emphasizes the multifaceted approach needed to understand and combat the belief in fake news.

Apuke and Omar [1] examine the factors influencing the sharing of fake news about COVID-19 on social media, using a Nigerian sample. Altruism emerged as the most significant predictor, with motivations related to information sharing, socialization, and information seeking also contributing to the spread of misinformation. The study concludes with implications for understanding the motivations behind fake news sharing, suggesting targeted interventions to reduce its spread.

Yang and Tian [60] focused into the third-person perception (TPP) regarding COVID-19 fake news susceptibility on social media. They find that communal engagement increases TPP, mediated by perceived knowledge and self-efficacy, but not by negative emotion. The study suggests that while social media can propagate fake news, it also fosters a belief among users in their own immunity to misinformation, highlighting the complex role of social media in the dissemination of fake news.

Wei et al. [61] Investigate the moderating role of social media literacy skills in the relationship between rational choice factors and fake news sharing behavior among Nigerian social media users. Trust in social media and status.

The proliferation of informal education through social media presents both unparalleled opportunities and significant challenges. On one hand, it democratizes access to knowledge, allowing lifelong learners from all walks of life to continuously engage with new information. This accessibility is particularly beneficial in an era where continuous learning and adaptation to new knowledge are crucial. However, the openness of these platforms also paves the way for the widespread dissemination of misinformation or fake news. The risk is more pronounced among those who may not belong to the digital native generation, such as older adults, who might lack the digital media literacy skills necessary to differentiate between authentic and fraudulent content. The issue of fake news is not just about the spread of misinformation but also about the potential it has to influence public opinion and behavior based on false premises. This scenario highlights the critical need for enhanced media literacy that includes the ability to critically assess the credibility and authenticity of online content. The integration of Artificial Intelligence (AI) tools, particularly machine learning and deep learning algorithms, in media literacy efforts presents a promising solution. These technologies can analyze vast amounts of data, identify patterns, and utilize sentiment analysis to gauge the authenticity and bias of the content. By using AI, a user alert system can be created to report potential misinformation, thereby acting as a digital literacy tool that empowers users to navigate the vast seas of information with a more discerning eye. Thus, while social media serves as a valuable educational resource, ensuring the reliability of the content consumed requires a concerted effort to foster digital literacy, especially among those not naturally versed in digital technologies. The use of AI and sentiment analysis offers a bridge across this digital divide, equipping users with the means to critically engage with content and become more resistant to the pitfalls of fake news. This approach not only protects individuals but also preserves the integrity of the digital information ecosystem, making it a safer space for educational purposes and informed discussions.

## 2.4. Motivation and research contribution

This work is driven by the growing global use of the internet and social media for sharing info. These digital places help us connect with people all over the world fast, but it is hard to know if what we read is true. So many lies and fake news spread on them easily. The urgency to address this issue is highlighted by the widespread dissemination of COVID-19-related misinformation, posing significant risks to public health. The research also stresses how important it is to stop false information online so many people get the COVID-19 vaccine. In this situation, sentiment analysis is very important because it allows us to analyze the information in real time. This helps stop fake information from spreading especially when talking about health or public info.

In concluding our discussion on sentiment analysis methodologies, it is noteworthy that Deep Learning has emerged as a prominent choice, surpassing other approaches like Machine Learning and traditional Lexicon-based methods. This is attributed to the superior accuracy demonstrated by Deep Learning when applied to extensive datasets with diverse values,

outperforming both ML and Lexicon-based techniques. The existing literature extensively explores sentiment analysis through various machine learning and deep learning algorithms. Each application yields distinct performance indicators; some studies report the use of logistic regression, while others employ Support Vector Machine or LSTM. Despite these findings, there is a clear need for more comprehensive research articles that apply a variety of learning methods to fully comprehend the comparative landscape between different algorithms, especially in the context of evolving deep learning techniques. In light of this, the contributions of our work can be summarized as follows: Deep Learning is recommended to replace lexicon-based methods and establish a platform for evaluating sentiment analysis algorithms in natural language processing (NLP) for textual feature extraction.

Consequently, the research contribution and the objective of this research is to employ five diverse machine learning classifiers, including Support Vector Machine, Logistic Regression, K-Nearest Neighbors, Decision Trees, and Random Forest, alongside three distinct deep learning techniques—CNN, LSTM, and GRU algorithms—to classify whether a Twitter (now X) post is authentic or not.

The article is organized into five sections. The upcoming section will focus into the methodology, explaining the details such as the dataset, pre-processing procedures, and the classification algorithms employed. Section 4 is dedicated to presenting the results and initiating discussions. This involves utilizing classification metrics and confusion matrices to provide a comprehensive analysis. Finally, the conclusion and recommendation section is included at the end of the article.

## 3. Methodology

The overall research methodology is illustrated in Fig 1, encompassing a systematic approach from problem identification to dataset development, pre-processing, classification, and subsequent implications. The problem statement underscores the rapid dissemination of COVID-19 information on platforms like Twitter, leading to misinformation, rumors, and conspiracy theories. This highlights the urgent need for real-time monitoring and community engagement to counter the severe consequences of false information on public health. Consequently, the proposed solution involves implementing fake news detection filters on social networks to curb the spread of misinformation. The fake news detection process entails several steps, elaborated upon herein.

### 3.1. Dataset

This study uses the "COVID Fake News Dataset" from kaggle.com. [https://www.kaggle.com/datasets/csmalarkodi/covid-fake-news-dataset The dataset, collected from news articles by scraping various websites and manually annotated by experts, was subsequently published on Kaggle for public access and research use. The dataset used for this work was retrieved on 28-08-2023.There are 6420 tweets, 3360 of which are authentic and 3060 fraudulent. The dataset is publicly available. The collection has three columns: the unique identification (ID), the tweet content, and the label, which distinguishes authentic from fraudulent tweets, like in the example shown in **Table 2** which is discussed briefly below:

- **ID**: This column contains unique identification numbers assigned to each entry in the dataset, allowing for easy referencing and identification of individual tweets.

- **Tweet:** This column comprises text data representing specific tweets related to the COVID-19 pandemic. Each row encapsulates the textual content of a particular tweet, ranging from updates, statistics, precautions, to opinions on the pandemic.

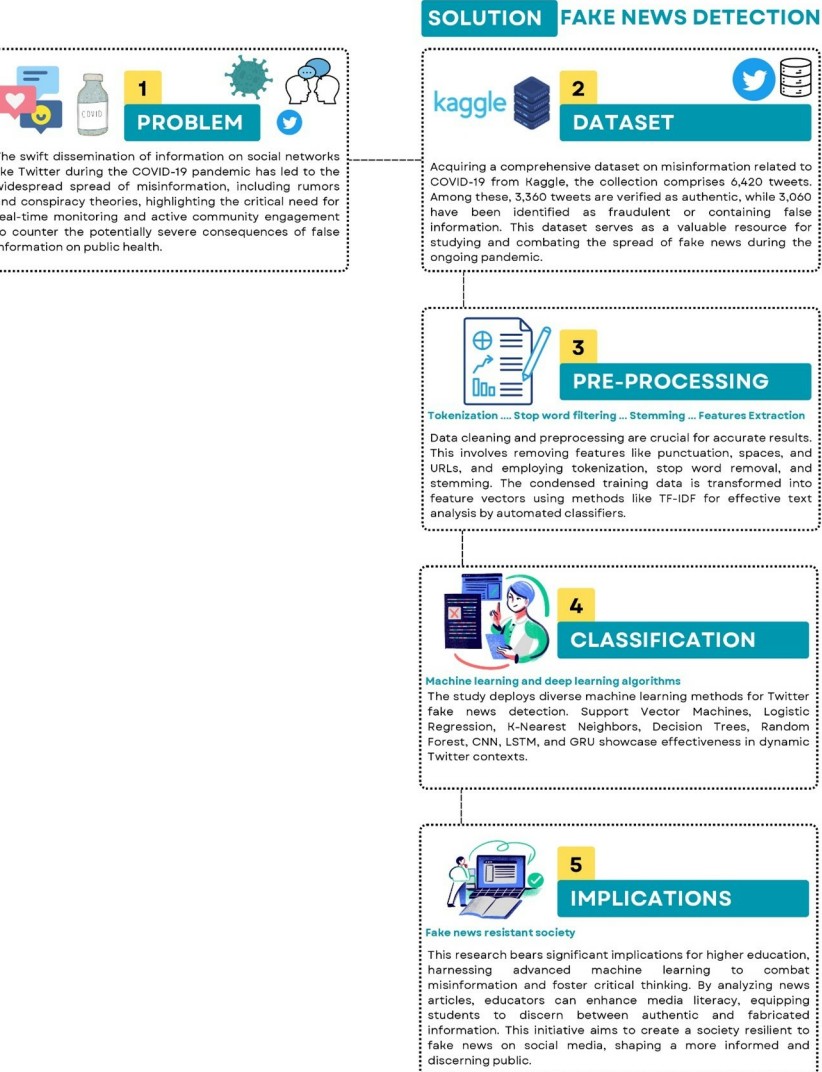

**Fig 1. Integral methodological diagram outlining the problem statement to the implications through the development of dataset, pre-processing, and classification.**

- **Label**: This column serves as a classifier, indicating the authenticity of the information contained in each tweet regarding COVID-19. Tweets are categorized as "real" when containing factual or authentic information and labeled as "fake" when potentially conveying misleading or false information pertaining to the pandemic.

In summary, Table 2 consists of columns representing unique identification numbers, the textual content of COVID-19-related tweets, and a classification label distinguishing between authentic ("real") and potentially misleading ("fake") information shared within the dataset. The dataset is labeled when its obtained from the recourse and no require any human annotation

### 3.2. Pre-processing

Once the data has been acquired, it is crucial to clean and preprocess it before proceeding with further processing. Utilizing thoroughly cleaned and preprocessed data is essential for

**Table 2. Sample data having columns with the information like ID, Tweet, and Label.** Adapted from: [https://www.kaggle.com/datasets/csmalarkodi/covid-fake-news-dataset].

| ID | Tweet | Label |
|---|---|---|
| 1 | The CDC currently reports 99031 deaths. In general the discrepancies in death counts between different sources are small and explicable. The death toll stands at roughly 100000 people today. | real |
| 2 | States reported 1121 deaths a small rise from last Tuesday. Southern states reported 640 of those deaths. https://t.co/YASGRTT4ux | real |
| 3 | Politically Correct Woman (Almost) Uses Pandemic as Excuse Not to Reuse Plastic Bag https://t.co/thF8GuNFPe #coronavirus #nashville | fake |
| 4 | #IndiaFightsCorona: We have 1524 #COVID testing laboratories in India and as on 25th August 2020 36827520 tests have been done: @ProfBhargava DG @ICMRDELHI #StaySafe #IndiaWillWin https://t.co/Yh3ZxknnhZ | real |
| 5 | Populous states can generate large case counts but if you look at the new cases per million today 9 smaller states are showing more cases per million than California or Texas: AL AR ID KS KY LA MS NV and SC. https://t.co/1pYW6cWRaS | real |
| 6 | Covid Act Now found "on average each person in Illinois with COVID-19 is infecting 1.11 other people. Data shows that the infection growth rate has declined over time this factors in the stay-at-home order and other restrictions put in place." https://t.co/hhigDd24fE | real |
| 7 | If you tested positive for #COVID19 and have no symptoms stay home and away from other people. Learn more about CDCâ€™s recommendations about when you can be around others after COVID-19 infection: https://t.co/z5kkXpqkYb. https://t.co/9PaMy0Rxaf | real |
| . | . | . |
| . | . | . |

obtaining accurate and favorable results [62]. In this specific procedure, various features such as punctuation marks, excessive spaces, and URLs are removed. Tokenization, the process of splitting a statement into smaller parts, either at the sentence or word level, is employed. Stop words, like "the," "for," "is," and "and" are excluded from the document as they do not contribute significantly to its meaning. The transformation of a term derived from its root word is known as "stemming." This involves converting terms like "needed" to "need," "eating" to "eat," and "cutting" to "cut." The training data has been condensed to a higher degree. Automated classifiers, which typically do not directly analyze text, require the conversion of text into a digital representation in the form of a vector. This digital vector is commonly referred to as a feature vector, and the conversion process is known as feature extraction. The TF-IDF (Term Frequency—Inverse Document Frequency) method [63] is widely used in the realm of feature extraction.

The pre-processed data is presented visually using both a histogram and a word cloud, as illustrated in Fig 2. In the context of sentiment analysis on Twitter (now X) data during the COVID-19 pandemic, two visualizations—a bar chart and a word cloud—offer valuable insights into the most frequently occurring words after pre-processing. The bar chart illustrates the frequency distribution of selected keywords. Notably, the term "COVID" dominates with a frequency of approximately 3500, followed by "cases" and "coronavirus," both hovering around 1800. Additionally, words such as "new," "people," "test," "deaths," "states," "total," "number," and "India" exhibit frequencies ranging from 500 to 1000. This quantitative representation underscores the prominence of specific terms in the dataset. Complementing the bar chart, the word cloud visually emphasizes the significance of certain terms. Words like "COVID," "new," "cases," "confirmed," "India," "coronavirus," "deaths," "vaccine," and "quarantine" stand out prominently. The varying font sizes in the word cloud directly correlate with the frequency of each term, providing an immediate visual cue to their importance. The discussion stemming from these visualizations could delve into the prevalent sentiments

**(a)**

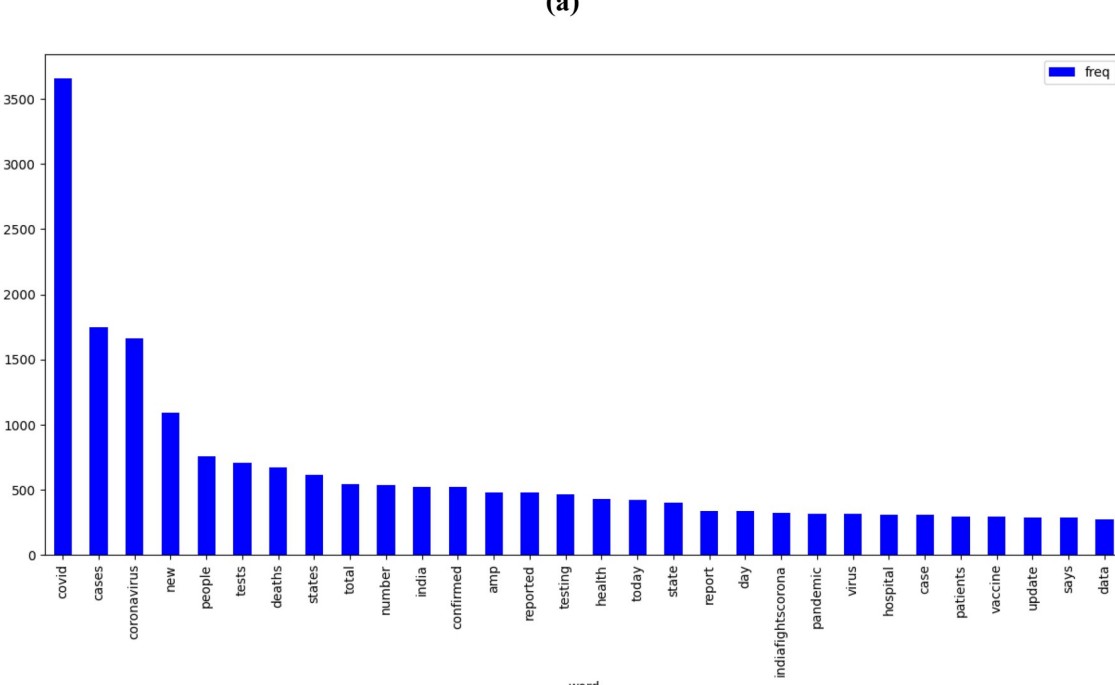

**(b)**

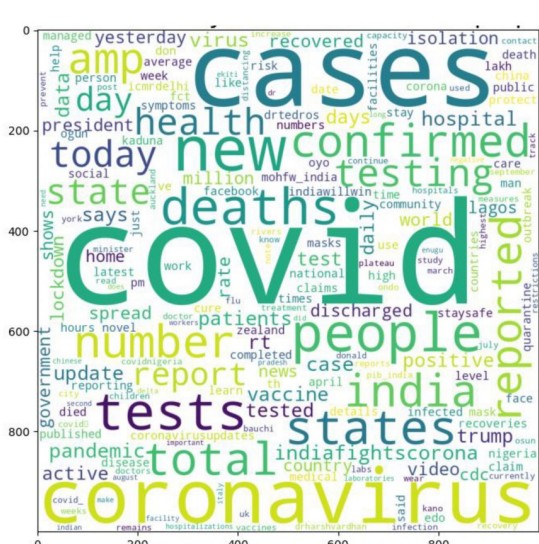

**Fig 2.** (a) Most frequently occurring words, and (b) word cloud, at post-processing.

expressed on Twitter (now X) during the pandemic. For instance, the high frequency of terms like "cases" and "deaths" may indicate a heightened awareness and discussion around the impact of the virus. Additionally, the prominence of "vaccine" suggests a significant focus on vaccination efforts and related sentiments. These visualizations offer a quantitative and qualitative understanding of the sentiments encapsulated in the Twitter (now X) data, aiding in a more nuanced interpretation of public discourse during the COVID-19 pandemic.

### 3.3. Data vectorization

The data vectorization step involves converting text which is input data into numerical vectors as most machine learning algorithms cannot process text directly. The transformation is achieved using the bag-of-words technique, treating text as a collection of words without considering grammar or order. TF-IDF is then used to assign scores to words, measuring their importance in documents. This process creates digital representations essential for algorithms to analyze textual data.

### 3.4. Classification

The dataset has two parts: training and testing. Training takes up 70% of the dataset, while testing and assessment take up 30%. The models used in this study undergo training with a specific dataset, constituting 70% of the total dataset. Testing is then conducted using a subset of data, specifically 30% of the entire dataset. Model evaluation relies on performance metrics, including the confusion matrix and classification rate. The analysis outcomes predict and classify the results as either "Real" or "Fake."

The following is a list of machine learning and deep learning algorithms, accompanied by an explanation of their implementation procedures.

**3.4.1. Support Vector Machines (SVM).** Support Vector Machines (SVM) can be used to address both linear and nonlinear problem domains, often yielding significantly improved outcomes in numerous real-world applications [44]. These machines partition data into distinct classes by utilizing a separating line or hyperplane. Nonlinear problems can be more effectively addressed through the use of kernels, as kernels facilitate the extraction of low-dimensional input space and its transformation into a higher-dimensional space. The SVM algorithm has the capability to perform intricate data transformations and effectively separate data into meaningful classes [64].

Defaultly, the Radial Basis Function (RBF) kernel is employed, with the regularization parameter C set to 1.0.

**3.4.2. Logistic Regression (LR).** Logistic Regression is a mathematical model that utilizes a matrix of variables to determine the weight of each variable. Based on this weight, the model predicts the classification of fake news related to COVID-19, represented in the form of a word matrix [44]. The deployment of a logistic regression (LR) classifier is contingent upon the presence of a binary dependent variable. In logistic regression, the relationship between independent and dependent variables is not linear. In the context of logistic regression, it is commonly observed that the variables are not normally distributed.

In logistic regression, the default values are set as follows: C = 1.0, solver = 'lbfgs', and max_iter = 100.

**3.4.3. K-Nearest Neighbor (KNN).** The K-nearest neighbors (KNN) algorithm has been used in this research. The K-Nearest Neighbors (KNN) algorithm is a machine learning (ML) technique that falls within the category of supervised ML classifiers. The utilization of the K-nearest neighbors (KNN) algorithm has the potential to address both regression and classification issue formulations. The sample is analyzed and classified to its nearest neighbor with the assistance of majority voting. The K-nearest neighbors (KNN) algorithm utilizes a distance matrix to identify the nearest neighbor with the least distance. The Euclidean distance, referred to as the distance metric, is utilized to estimate the distance between training instances and testing examples [44]. The prediction and classification of a new data set is achieved by utilizing the nearest training instances to produce a certain value. The K-Nearest Neighbors (KNN) algorithm is commonly utilized in scenarios that demand high prediction accuracy, as it has demonstrated the ability to generate very accurate predictions.

In KNN, the default value for the number of neighbors is set to 5.

**3.4.4. Decision Trees (DT).** Its popularity makes the decision tree algorithm popular in data mining. The hierarchical structure with several possibilities qualifies as a "model of choices." Decision trees mark class traits on leaves and describe them in inner nodes. Decision trees are popular since they're simple. The Decision Tree (DT) method selects the best attribute, which provides valuable categorization information. The process stops when all leaf nodes are pure or when classification is no longer needed [65].

In decision tree, the default values are utilized as follows: criterion = 'gini', max_depth = None, and min_samples_leaf = 1.

**3.4.5. Random Forest (RF).** The Random Forest algorithm had been utilized in this research It is an accumulation technique that operates by combining the principles of bagging and spaces [66]. The Random Forest algorithm generates a collection of Decision Trees using a given training dataset [67]. Following the aggregation of votes, the label is determined based on inputs from many decision trees. Random Forest has a wide range of applications in various domains, including but not limited to distant sensing and drug discovery [14].

In random forest, default values are applied as follows: n_estimators = 100, criterion = 'gini', max_features = 'auto', and random_state = 42.

**3.4.6. Convolutional Neural Network (CNN).** This classifier uses deep learning. Due to its unique construction, the Multilayered Perceptron becomes more complicated when parameters are added. Scholars use deep learning to reduce such complexities [68]. CNN is a popular deep learning classifier. It is widely used in computer vision, image processing, and pattern recognition research. Input reaches deeper layers, culminating in conceptual features. CNN is known for requiring less data preprocessing than other computer vision and image processing classifiers. CNN has many benefits, but it cannot recognize transient data.

The activation function ReLU is employed with the Adam optimizer for CNN. The configuration includes 10 epochs, a batch size of 32, and a validation split of 0.1. Other parameters set are embedding_dim = 50, filter_size = 128, kernel_size = 3, pool_size = 3, dense_units = 256, and dropout_rate = 0.2.

**3.4.7. Long Short-Term Memory (LSTM).** This model is a deep learning classifier. Feedforward neural networks lack feedback connections, limiting them. Long Short-Term Memory (LSTM) networks with feedforward and feedback links can solve this problem [44]. Cellular mechanisms decide the value of each employee termination, while remaining gateways handle data and regulate communication. Backpropagation can cause disappearing gradients in Recurrent Neural Networks (RNNs), which update weights depending on gradients. The Long Short-Term Memory (LSTM) model solves such problems by using internal gates to govern information flow within and between cells. Voice recognition, text generation, and voice analysis are frequent uses.

The Adam optimizer is utilized with 10 epochs, a validation size of 0.1, and a batch size of 32 for the specified configuration. Other parameters include embedding_dim = 50, lstm_units = 100, dense_units = 256, and dropout_rate = 0.2.

**3.4.8. Gated Recurrent Units (GRU).** The approach outperforms Long Short-Term Memory in music signal modeling, speech recognition, and natural language processing [69]. In small datasets, the GRU model performs well and produces accurate results. Also, it mitigates gradient vanishing in the reset and update gates dilemma. This system's ability to retain and use unnecessary data, determine its importance, and output it is its main benefit. GRU may produce exact results on large, complex datasets when trained extensively. Researchers use Gated Recurrent Units (GRU) in many practical applications.

For GRU, the configuration includes 10 epochs, a batch size of 20, and the use of the Adam optimizer with a validation split of 0.1. The specified parameters are embedding_dim = 50, lstm_units = 64, gru_units = 32, dense_units = 128, and dropout_rate = 0.2.

## 4. Results and discussion

This study employs data and text mining to analyze the emotional impact of COVID-19 pandemic misinformation. Nine classifiers, including LSTM, BiGRU, Random Forest, KNN, Logistic Regression, Decision Trees, SVM, 1d-CNN, and Naive Bayes, were implemented. The experiments were conducted on Intel(R) Core(TM) i5-6300U CPUs @ 2.40GHz and 2.50 GHz with 16 GB RAM. The study utilized Colab Python 3.9 on Windows 10 64-bit for result generation.

Classification rates and confusion matrices are employed for training and evaluating models. A dataset containing spurious COVID-19 information is utilized to assess the classifier's performance. The objective is to predict the emotive categorization of each fake news item related to COVID-19 in this dataset. Several machine learning methods, detailed in Table 3, were utilized in this study.

Performance is gauged through accuracy, precision, recall, and F1 score matrices. Accuracy in classification is a metric that measures the overall correctness of a model's predictions across all classes. The accuracy of a model is calculated by dividing the number of correct predictions with the total number of predictions. Mathematically, it can be written as:

$$Accuracy = \frac{Number\ of\ correct\ predictions}{Total\ number\ of\ predictions} \tag{1}$$

Precision in classification is a metric that measures the accuracy of the positive predictions made by a classification model. Specifically, precision is the ratio of true positive predictions to the total number of positive predictions made by the model, mathematically written as:

$$Precision = \frac{True\ positives}{True\ positives + False\ positives} \tag{2}$$

Recall is the ratio of true positives to the sum of true positives and false negatives, and mathematically, it can be written as:

$$Recall = \frac{True\ positives}{True\ positives + False\ negatives} \tag{3}$$

In summary, precision and recall are crucial elements in any classification problem.

The F1 score is a metric that combines precision and recall into a single value to provide a more comprehensive assessment of a classification model's performance. It is particularly useful when dealing with imbalanced datasets where one class may be significantly more prevalent than the other. Mathematically, it can be written as:

$$F1\ score = 2 \times \frac{Precision \times Recall}{Precision + Recall} \tag{4}$$

This data is further elaborated in Table 3(A). The performance indices for the machine learning algorithms are presented in Table 3(A), while those for the deep learning algorithms are displayed in Table 4(A).

Tables 3(A) and 4(A) provide a comprehensive overview of the statistical performance indicators, showcasing the overall accuracy, precision, recall, and F1-score for the machine learning and deep learning algorithms, respectively. To delve deeper into the evaluation metrics, it's

**Table 3.** Statistical performance indicators for machine learning algorithms in classification can be evaluated through (a) overall metrics and (b) classification based on macro and weighted indicators.

| (a) | | | | |
|---|---|---|---|---|
| Model | Accuracy | Precision | Recall | F1-score |
| SVM | 0.89 | 0.88 | 0.91 | 0.90 |
| Random Forest | 0.89 | 0.87 | 0.92 | 0.90 |
| Decision Tree | 0.85 | 0.84 | 0.87 | 0.85 |
| Logistic Regression | 0.89 | 0.87 | 0.92 | 0.89 |
| KNN | 0.73 | 0.96 | 0.50 | 0.66 |

| (b) | | | | | | |
|---|---|---|---|---|---|---|
| Metrics | Average | SVM | Logistic regression | Random forest | KNN | Decision tree |
| Precision | Macro | 0.90 | 0.89 | 0.89 | 0.80 | 0.85 |
| | Weighted | 0.90 | 0.89 | 0.89 | 0.81 | 0.85 |
| Recall | Macro | 0.89 | 0.89 | 0.89 | 0.74 | 0.85 |
| | Weighted | 0.89 | 0.89 | 0.89 | 0.73 | 0.85 |
| F1 score | Macro | 0.89 | 0.89 | 0.89 | 0.72 | 0.85 |
| | Weighted | 0.89 | 0.89 | 0.89 | 0.72 | 0.85 |

essential to consider both macro and weighted precision, recall, and F1 score, as detailed in Tables 3(B) and 4(B). Macro precision, recall, and F1 score calculate the unweighted average across all classes, treating each class equally. On the other hand, weighted precision, recall, and F1 score provide an average that considers the class distribution, assigning more significance to larger classes. This evaluation is crucial in scenarios where class imbalances exist, offering a more holistic understanding of a model's performance. The consideration of macro and weighted metrics contributes to a more robust assessment, allowing a better interpretation of the algorithms' effectiveness across various classes. This approach is particularly beneficial in real-world applications where class distribution imbalances are common. In addition to providing a clearer picture of a model's capabilities, it aids in identifying areas for improvement and refining strategies for enhanced algorithmic performance.

While observing Tables 3(A) and 4(A) together, among the machine learning models, Support Vector Machines (SVM), Random Forest, and Logistic Regression demonstrated high accuracy, all achieving an impressive 0.89. However, their performance varies

**Table 4.** Statistical performance indicators for deep learning algorithms in classification can be evaluated through (a) overall metrics and (b) classification based on macro and weighted indicators.

| (a) | | | | |
|---|---|---|---|---|
| Model | Accuracy | Precision | Recall | F1-score |
| LSTM | 0.90 | 0.93 | 0.88 | 0.90 |
| BiGRU | 0.91 | 0.90 | 0.93 | 0.92 |
| CNN-1D | 0.71 | 1.00 | 0.43 | 0.60 |

| (b) | | | |
|---|---|---|---|
| Metrics | Average | LSTM | BiGRU | CNN-1D |
| Precision | Macro | 0.90 | 0.91 | 0.81 |
| | Weighted | 0.90 | 0.91 | 0.82 |
| Recall | Macro | 0.90 | 0.91 | 0.72 |
| | Weighted | 0.90 | 0.91 | 0.71 |
| F1 score | Macro | 0.90 | 0.91 | 0.69 |
| | Weighted | 0.90 | 0.91 | 0.68 |

concerning precision, recall, and F1-score. Random Forest emerged as the leader with the highest recall (0.92), indicating its capability to effectively capture true positive instances. SVM closely followed with balanced precision (0.88) and recall (0.91), resulting in a harmonious F1-score of 0.90. Logistic Regression also exhibited competitive metrics, particularly in precision and recall. On the other hand, the K-Nearest Neighbors (KNN) model, while achieving a high precision of 0.96, struggled with recall (0.50), resulting in a comparatively lower F1-score of 0.66. This suggests that KNN excelled in correctly identifying positive instances but may have missed a substantial number of actual positive cases. Turning to the deep learning models, the BiGRU model stands out with an accuracy of 0.91 and well-balanced precision, recall, and F1-score (0.90, 0.93, and 0.92, respectively). This indicates its robust performance in correctly classifying both positive and negative instances. LSTM also performed well with an accuracy of 0.90, emphasizing its effectiveness in sentiment analysis. Conversely, the CNN-1D model demonstrated lower performance across all metrics, particularly in recall (0.43), suggesting challenges in capturing true positive instances. While achieving high precision (1.00), it came at the cost of a lower recall, resulting in a lower F1-score (0.60). In summary, these results highlight the varying strengths and weaknesses of each model. The choice of a suitable model depends on the specific requirements of the sentiment analysis task, considering factors such as the importance of precision versus recall. Additionally, the findings underscore the importance of a comprehensive evaluation using multiple metrics to gain a better understanding of model performance.

While observing Tables 3(B) and 4(B) together, across SVM, Logistic Regression, and Decision Tree models, the precision, recall, and F1 scores consistently average around 0.89, both in macro and weighted evaluations. This suggests a balanced performance across all classes for these models. Notably, the Random Forest model exhibits slightly lower performance in terms of macro F1 score (0.72), indicating potential challenges in achieving a harmonious trade-off between precision and recall. KNN stands out with exceptionally high precision (both macro and weighted) at 0.96. However, this high precision comes at the cost of lower recall metrics, resulting in an overall lower F1 score of 0.66. This indicates that while KNN excels in correctly identifying positive instances, it may struggle to capture all actual positive cases. Turning to the deep learning models, both LSTM and BiGRU demonstrate robust overall performance, with macro and weighted precision, recall, and F1 scores averaging around 0.91. These models showcase a well-balanced ability to correctly classify both positive and negative instances, indicating strong sentiment analysis capabilities. In contrast, the CNN-1D model achieves high precision (both macro and weighted) but lags in recall metrics, resulting in a lower F1 score of 0.69. This suggests potential challenges in capturing true positive instances, indicating a more conservative approach in labeling instances as positive. In summary, the macro and weighted averages portray a consistent outperformance of deep learning models (LSTM, BiGRU, and CNN-1D) compared to machine learning models. The breakdown of precision, recall, and F1 scores provides valuable insights into the models' specific strengths and weaknesses, guiding the choice of models based on the specific requirements of sentiment analysis tasks.

The assessment of classifier performance is enhanced using a confusion matrix, offering a detailed breakdown of various performance parameters. In Figs 3 and 4, the outcomes are presented, focusing on machine learning and deep learning, respectively. The confusion matrix is a 2x2 matrix with predicted values (0 and 1) represented in columns and actual values (0 and 1) in rows. Instances where the actual class is 0, and the model correctly predicts 0, are termed True Negatives (TN). Conversely, instances where the actual class is 1, and the model correctly predicts it as 1, are known as True Positives (TP). False Negatives (FN) occur when the prediction is 0, but the actual class is 1, signifying instances where the model incorrectly predicts 0.

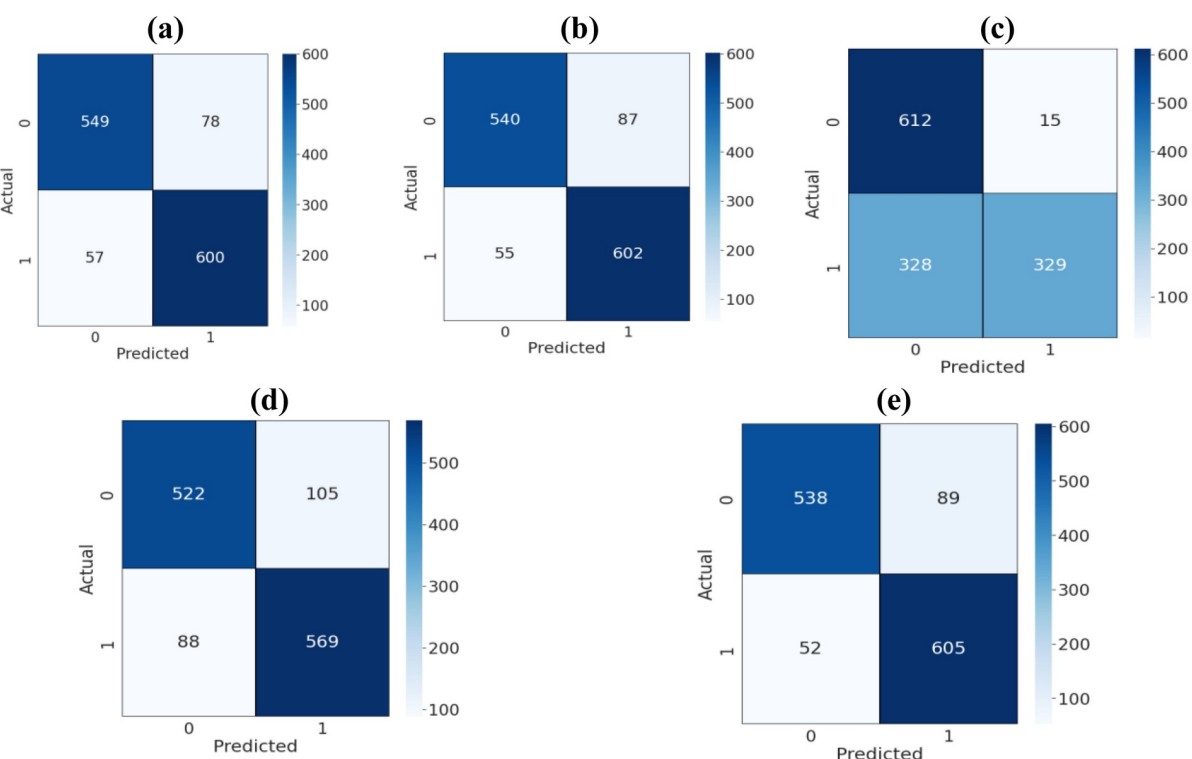

**Fig 3.** Confusion matrix for (a) SVM, (b) logistic regression, (c) K-nearest neighbor, (d) decision trees, and (e) random forest.

On the other hand, False Positives (FP) are instances where the actual class is 0, but the model incorrectly predicts it as 1. This comprehensive breakdown allows for a nuanced evaluation of the model's predictive accuracy and error types.

The confusion matrices provide a comprehensive view of the classification performance for various machine learning models. The Support Vector Machine (SVM) and Logistic Regression models exhibit balanced predictions, showcasing a notable equilibrium between true positives and true negatives. This suggests a well-rounded performance in both precision and

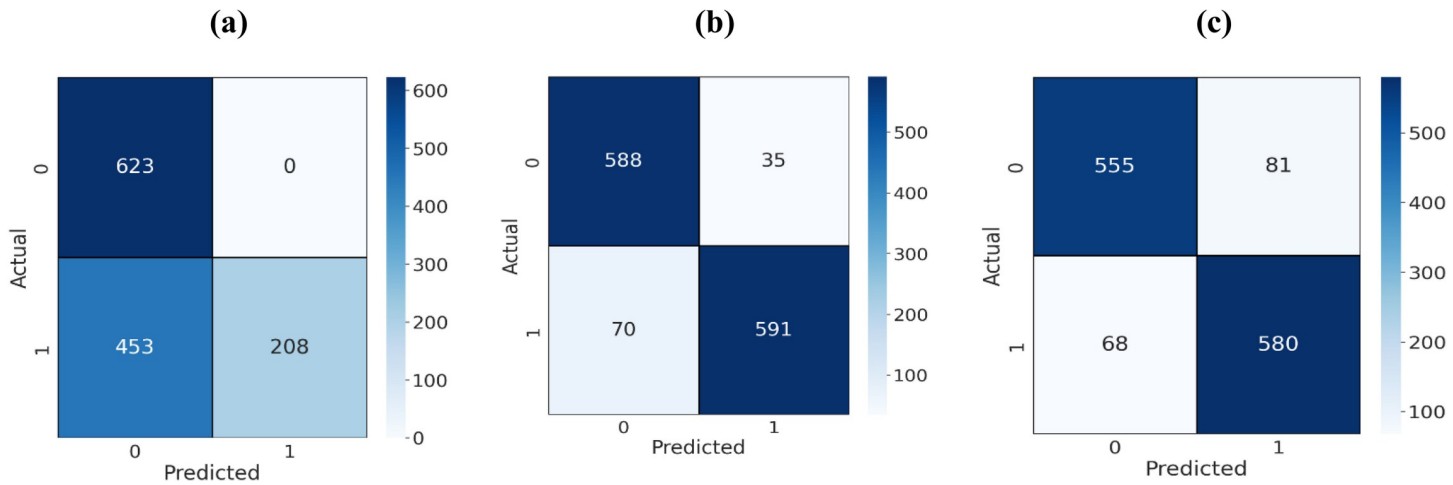

**Fig 4.** Confusion matrix for (a) CNN, (b) LSTM, and (c) GRU.

recall. On the other hand, the K-Nearest Neighbor model demonstrates a higher number of true positives, indicating its proficiency in correctly identifying positive instances. However, it also incurs a notable number of false negatives, implying a potential room for improvement in recall. Moving on to Decision Trees and Random Forest, both models display a commendable balance with respectable true positive and true negative values. The Convolutional Neural Network (CNN) exhibits an impressive performance with no false positives, although it could benefit from an increase in true positives for a more robust classification of positive instances. Lastly, Long Short-Term Memory (LSTM) and Gated Recurrent Unit (GRU) modelsfu also show balanced performance with a minimal number of false predictions, affirming their efficacy in capturing temporal dependencies in the data. Overall, each model demonstrates strengths and areas for refinement, emphasizing the importance of selecting the appropriate algorithm based on the specific characteristics of the dataset and the desired trade-offs between precision and recall.

## 4.1. Comparison with previous studies

In the rapidly evolving landscape of the COVID-19 pandemic, the detection and mitigation of misinformation have This comparison aims to highlight the strengths of Our proposed study in achieving superior results in the classification of COVID-19-related news compared to [70, 71] leverages a dataset amalgamated from various social media and news sources, focusing on the preprocessing steps, tokenization, and feature selection to enhance the efficiency of the classification process. The utilization of several state-of-the-art machine learning algorithms demonstrates a meticulous approach to model selection. Notably, the random forest classifier attains an impressive accuracy of 88.50%. While employing a diverse set of classifiers, including both machine learning and deep learning models, the BiGRU deep learning classifier is identified as the most effective, achieving an accuracy of 0.91. The comparison between these studies reveals that random forest classifier outperforms the classifiers in in terms of accuracy. While these studies address the critical issue of misinformation related to COVID-19, the distinctive strength of studies lies in its meticulous feature selection process and the careful consideration of various machine learning algorithms. The 88.50% accuracy achieved by study [70] random forest classifier signifies a robust performance in distinguishing between authentic and misleading COVID-19-related news.

## 4.2. Limitation and challenges

Detecting COVID-19 misinformation is challenged by the complexities inherent in natural language processing, requiring navigation through intricate nuances, varied contextual elements, and subjective aspects for accurate identification. The temporal dimension is crucial; conducting analyses within specific lockdown phases risks producing irrelevant findings due to evolving public sentiments. Neglecting emoticons and hashtags poses a significant challenge, compromising sentiment interpretation and classifier efficacy, especially in contexts involving sarcasm or contradictory expressions. Further, limited reliance on lexicons restricts the depth needed to identify evolving narratives of COVID-19-related fake news, potentially hindering the understanding of nuanced misinformation during the pandemic. Addressing these challenges mandates a comprehensive strategy encompassing language intricacies, adapting to temporal changes, integrating emoticons and hashtags, and broadening lexicon use. To achieve heightened accuracy in detecting COVID-19-related fake news, an approach encompassing the multifaceted information landscape during this unprecedented global crisis is imperative.

## 5. Conclusion and recommendation

In this comprehensive study, a diverse set of six Machine Learning and three Deep Learning classifiers were employed to analyze misinformation surrounding the COVID-19 epidemic. The analysis encompassed well-established classifiers such as Support Vector Machine, Random Forest, Decision Trees, K-Nearest Neighbor, and Logistic Regression, alongside advanced Deep Learning models, including CNN, LSTM, and BiGRU. The search of the optimal sentiment analysis model led to rigorous experimentation using false news data from a reputable source, with evaluation of recall, precision, accuracy, and F1 score for each classifier.

The findings highlight the best performance of the BiGRU deep learning classifier, showcasing exceptional accuracy and efficiency. The reported metrics, including an accuracy of 0.91, precision of 0.90, recall of 0.93, and F1-score of 0.92, underscore its effectiveness in accurately categorizing sentiments. For this algorithm, true negatives and true positives numbered 555 and 580, respectively, while false negatives and false positives were limited to 81 and 68, respectively.

The ramifications of this research reach beyond academic spheres, aiming to contribute to the creation of a society resilient to the dissemination of fake news. This study is of considerable significance for higher education and lifelong learners, illuminating the potential of advanced machine learning in the persistent struggle against misinformation. By nurturing critical thinking skills among students, the research empowers educators and institutions to deploy these methods effectively in the analysis and classification of news articles. Consequently, this facilitates the development of tools and curricula geared towards enhancing media literacy and information validation. The cultivation of such skills is of paramount importance, particularly in the context of the COVID-19 pandemic, equipping individuals with the discernment needed to differentiate between authentic and fabricated information. Through endeavors like this, we contribute to building a society fortified against the perils of fake news, fostering increased literacy and resilience among its members.

In future studies examining COVID-19 misinformation, the utilization of advanced Transformer-based architectures and larger language models, such as those within the GPT (Generative Pre-trained Transformer) series, like GPT, T5, and LLM, holds considerable promise for gaining a deeper understanding of complex linguistic patterns. By using these sophisticated models, researchers can effectively analyze and interpret intricate textual data related to misinformation surrounding the COVID-19 pandemic. Expanding the dataset's scope across diverse sources and domains will bolster the model's resilience and flexibility. The integration of interpretability techniques like Shapley and LIME will be crucial, allowing for clearer explanations and validation of the model's decision-making rationale. These approaches contribute significantly to understanding how the model processes information, fostering trust and transparency in decision-making. This convergence of advancements is anticipated to significantly transform the identification and comprehension of COVID-19-related fake news, leading to more dependable and informed decisions amid the dynamic information landscape.

## Supporting information

**S1 File.**
(DOCX)

## Author Contributions

**Conceptualization:** Muhammad Tayyab Zamir, Fida Ullah, Alexander Gelbukh.

**Data curation:** Muhammad Tayyab Zamir, Fida Ullah, Rasikh Tariq.

**Formal analysis:** Muhammad Tayyab Zamir, Fida Ullah, Rasikh Tariq, Muhammad Arif.

**Funding acquisition:** Muhammad Tayyab Zamir, Fida Ullah, Rasikh Tariq.

**Investigation:** Muhammad Tayyab Zamir, Fida Ullah, Rasikh Tariq, Waqas Haider Bangyal, Muhammad Arif, Alexander Gelbukh.

**Methodology:** Muhammad Tayyab Zamir, Fida Ullah, Rasikh Tariq, Waqas Haider Bangyal, Muhammad Arif.

**Project administration:** Muhammad Tayyab Zamir, Fida Ullah.

**Resources:** Muhammad Tayyab Zamir, Fida Ullah, Rasikh Tariq.

**Software:** Muhammad Tayyab Zamir, Fida Ullah.

**Supervision:** Muhammad Tayyab Zamir, Rasikh Tariq, Alexander Gelbukh.

**Validation:** Muhammad Tayyab Zamir, Fida Ullah.

**Visualization:** Muhammad Tayyab Zamir, Fida Ullah, Alexander Gelbukh.

**Writing – original draft:** Muhammad Tayyab Zamir, Fida Ullah, Rasikh Tariq.

**Writing – review & editing:** Rasikh Tariq, Waqas Haider Bangyal, Alexander Gelbukh.

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
