## [Decision Letter · Decision Letter 0]

28 Dec 2023

PONE-D-23-40611Machine and deep learning algorithms for sentiment analysis during COVID-19: a vision to create fake news resistant societyPLOS ONE

Dear Dr. Tariq,

Thank you for submitting your manuscript to PLOS ONE. After careful consideration, we feel that it has merit but does not fully meet PLOS ONE’s publication criteria as it currently stands. Therefore, we invite you to submit a revised version of the manuscript that addresses the points raised during the review process.

We look forward to receiving your revised manuscript.

Kind regards,

Fredrick Ishengoma

Academic Editor

PLOS ONE

Journal Requirements:

   "The authors would like to thanks the financial support from Tecnologico de Monterrey through the “Challenge-Based Research Funding Program 2022”. Project ID # I003 - IFE001 - C2-T3 – T"

4. In the online submission form, you indicated that "Data can be shared on request".

Reviewers' comments:

Reviewer's Responses to Questions

**Comments to the Author**

1. Is the manuscript technically sound, and do the data support the conclusions?

Reviewer #1: Partly

Reviewer #2: No

2. Has the statistical analysis been performed appropriately and rigorously? 

Reviewer #1: No

Reviewer #2: Yes

3. Have the authors made all data underlying the findings in their manuscript fully available?

Reviewer #1: No

Reviewer #2: Yes

4. Is the manuscript presented in an intelligible fashion and written in standard English?

Reviewer #1: Yes

Reviewer #2: Yes

5. Review Comments to the Author

Reviewer #1: 1. Dr. Y. Sun, North China Electric Power University - Beijing Campus: North China Electric Power University, China, sunyb@ncepu.edu.cn

2. Dr. Q. Fan, Chinese Academy of Sciences Chongqing Institute of Green and Intelligent Technology, China, fanqh@lzb.ac.cn.

3. Dr. Y. Liu, East China University of Technology, China, Walton_liu@163.com

The abstracts introductory portion is too long. Readers expect to see more detail of the methodology, results, and conclusion in the abstract. The abstract need to be greatly improved. The abstract does not show that the authors achieved much as there is no numerical justification to back the author’s claims or results of comparative analysis to show superior performance.

1. In the introduction, the authors should explain why they did it (motivation) discussing the possible outcome. Readers are primarily interested in the motivation and outcome of your research. Therefore, a good introduction should contain:

a. What is the problem to be solved?

b. Are there any existing solutions?

c. Which is the best?

d. What is the main limitation of the best and existing approaches?

e. What do you hope to change or propose to make it better?

f. How is the paper structured?

2. Please clearly highlight how your work advances the field from the present state of knowledge and you should provide a clear justification for your work which should be stated at the end of literature review/ related works. The impact or advancement of the work can also appear in the conclusion.

3. The authors mentioned feature extraction but have not presented this stage in their work. A block diagram or flowchart of the steps would be helpful. The authors should look for recent works on feature extraction to cite. An example of such recent literature which the authors can consult amongst others is:

-Feature Extraction: A Survey of the Types, Techniques, Applications, 5th IEEE International Conference on Signal Processing and Communication (ICSC), Noida, India, pp. 158-164 (2019). DOI: 10.1109/ICSC45622.2019.8938371

4. Authors should use a flowchart to illustrate the Procedure to Produce the FLDI index. This will help readers to understand the procedure better.

5. Related works section is not sufficient. The authors should improve on this section as they have left many papers out. Normally, it’s the gaps in work of others that the authors are expected to fill.Therefore, at the end of your review section state the problems in this field with appropriate reference and tell readers which one your work addresses.

The authors should consult and cite:

(i) Detection and Classification of COVID-19 Disease from X-ray Images Using Convolutional Neural Networks and Histogram of Oriented Gradients. Biomedical Signal Processing and Control, 103530, Vol. 74, pp. 1-11, 2022. DOI: 10.1016/j.bspc.2022.103530

(ii) Prediction of COVID-19 Outbreak with Current Substantiation Using Machine Learning Algorithms. Intelligent Interactive Multimedia Systems for e-Healthcare Applications. Springer, Singapore, 2022. https://doi.org/10.1007/978-981-16-6542-4_10

(iii) CIA-CVD: cloud based image analysis for COVID-19 vaccination distribution. Journal of Cloud Computing. Vol. 12, 163. DOI: 10.1186/s13677-023-00539-y

(iv) PETLFC: Parallel ensemble transfer learning based framework for COVID-19 differentiation and prediction using deep convolutional neural network models. Multimedia Tools and Applications. DOI: 10.1007/s11042-023-16084-4

(v) X-Ray image-based COVID-19 detection using deep learning. Multimedia Tools and Applications, Vol. 82, pp. 44507–44525. DOI: 10.1007/s11042-023-15389-8

(vi) Detection of Corona Virus Disease Using a Novel Machine Learning Approach. 2021 International Conference on Decision Aid Sciences and Application (DASA), pp. 587-590, 2021. DOI: 10.1109/DASA53625.2021.9682267.

6. Most of the figures in this paper are not clear enough. The authors should endeavour to change them. E.g. Figs. 1-4.

7. The authors need to discuss the results in Tables 1 and 2 better. The reason why the proposed technique performs better has not been explained.

8. There is no comparison of results with the existing works in this paper. This should be added for readers to see how your proposed method performs relative to other works.

9. It would be good for the authors to discuss other methods that different authors have used to take fake news.

Authors should read and cite:

- Detection of fake news and hate speech for Ethiopian languages: a systematic review of the approaches. Journal of Big Data, pp. 1-17. DOI: 10.1186/s40537-022-00619-x

10. The authors should structure the paper into abstract, introduction, literature review/related works, methodology, results and discussion, and conclusion.

11. I was hoping to see more results and discussion as more results could be presented to make the work much appreciable. The authors are encouraged to reduce the plagiarism of the paper.

12. The Limitations of the proposed study need to be discussed before conclusion.

13. Some of the challenges encountered during the course of the study can be highlighted and future recommendations can be added at the end of the conclusion. Retitle conclusion as conclusion and recommendation.

14. The results and discussion section is very weak. The authors should endeavor to improve on this section. In the section of selection of local minima, what criteria did the authors use. Also what priors did the authors consider? What is the minimum and the maximum values? If these are suitable, do they work for different types of images or just the images under consideration?

15. Lastly, no comparison of results was presented with other state-of-the-art methods which have used machine learning techniques

Reviewer #2: Overall this is very interesting work in ML and DL based on Sentiment analysis, but the overall presentation of the work needs a major improvement. The current manuscript needs significant improvements. Some of the improvements need to be done in this manuscript are as follows:

1. The Motivation and Contribution Subsection needs to be added in the Introduction Section. The major contribution and novelty of the work need to be briefly discussed.

2. A separate Section Literature Review needs to be added and a recent state-of-the-art scheme needs to be discussed.

3. Various ML and DL-based approaches and techniques need to be discussed in in New Section Background Detail.

4. Data Labeling and vectorization steps are considered in this work or not? Proper discussion is needed, as these are the major steps in sentiment analysis. For reference, the author may refer to the following work:Gupta, P., Kumar, S., Suman, R. R., & Kumar, V. (2020). Sentiment analysis of lockdown in india during covid-19: A case study on twitter. IEEE Transactions on Computational Social Systems, 8(4), 992-1002.

Jayasurya, G. G., Kumar, S., Singh, B. K., & Kumar, V. (2021). Analysis of public sentiment on COVID-19 vaccination using twitter. IEEE Transactions on Computational Social Systems, 9(4), 1101-1111.

5. In section IV the proper equation is needed for accuracy, recall, precision, and F1 score.  Every variable used in the equation needs proper discussion.

6. In Table 3,  what is the random state for the ML techniques? All these values change with the random state. It is recommended to check the result with varying random states or perform k-fold cross-validation. 

7. Further the result of this work needs to be compared with some of the recent state-of-the-art work.

8. What is the novelty of this work? There are several works in which this type of work is reported. So, it is recommended to add the novelty of the work in the motivation and contribution section.

6. PLOS authors have the option to publish the peer review history of their article (what does this mean?). If published, this will include your full peer review and any attached files.

Reviewer #1: No

Reviewer #2: **Yes: **sanjay kumar

---

## [Author Response · Author response to Decision Letter 0]

26 Jan 2024

The Response To Reviewer Comments is upload in the Attach Files Section.

---

## [Decision Letter · Decision Letter 1]

22 Mar 2024

PONE-D-23-40611R1Machine and deep learning algorithms for sentiment analysis during COVID-19: a vision to create fake news resistant societyPLOS ONE

Dear Dr. Tariq,

Thank you for submitting your manuscript to PLOS ONE. After careful consideration, we feel that it has merit but does not fully meet PLOS ONE’s publication criteria as it currently stands. Therefore, we invite you to submit a revised version of the manuscript that addresses the points raised during the review process.

We look forward to receiving your revised manuscript.

Kind regards,

Fredrick Romanus Ishengoma

Academic Editor

PLOS ONE

Journal Requirements:

Reviewers' comments:

Reviewer's Responses to Questions

**Comments to the Author**

1. If the authors have adequately addressed your comments raised in a previous round of review and you feel that this manuscript is now acceptable for publication, you may indicate that here to bypass the “Comments to the Author” section, enter your conflict of interest statement in the “Confidential to Editor” section, and submit your "Accept" recommendation.

Reviewer #2: All comments have been addressed

Reviewer #3: (No Response)

Reviewer #4: All comments have been addressed

2. Is the manuscript technically sound, and do the data support the conclusions?

Reviewer #2: Yes

Reviewer #3: Partly

Reviewer #4: Yes

3. Has the statistical analysis been performed appropriately and rigorously? 

Reviewer #2: Yes

Reviewer #3: Yes

Reviewer #4: Yes

4. Have the authors made all data underlying the findings in their manuscript fully available?

Reviewer #2: Yes

Reviewer #3: Yes

Reviewer #4: Yes

5. Is the manuscript presented in an intelligible fashion and written in standard English?

Reviewer #2: Yes

Reviewer #3: Yes

Reviewer #4: Yes

6. Review Comments to the Author

Reviewer #2: All of the suggested points by the reviewers are positively addressed by the authors. So, I will recommend to accept this manuscript

Reviewer #3: I appreciated the efforts made by the authors to compare the methods in sentiment analysis related to supervised ML and DL. Since this is the revised version, it is not fair to further point out that the comparison between SVM/RF etc. vs. CNN, LSTM, GRU addresses the state-of-the-art technique like LLMs. A series of recent research shows that some of these fine-tuned LLMs or even zero-short learning using GPT outperform traditional methods. This paper should have improved greatly if they can further include these methods.

Reviewer #4: This paper introduces a technique for detecting misinformation related to the COVID-19 pandemic. The experimental results indicate that the BiGRU deep learning classifier demonstrates a higher accuracy and efficiency than six machine learning and two deep learning classifiers. In general, the paper is well written and easy to understand. The reviewer believe that this paper can be accepted for publication, however, I do have some comments, in particular:

• Some of key publications in this field are missing in the introduction section. It is suggested to add them in.

• Some references are not formatted correctly.

7. PLOS authors have the option to publish the peer review history of their article (what does this mean?). If published, this will include your full peer review and any attached files.

Reviewer #2: No

Reviewer #3: No

Reviewer #4: No

---

## [Decision Letter · Decision Letter 2]

7 Jul 2024

PONE-D-23-40611R2Machine and deep learning algorithms for sentiment analysis during COVID-19: a vision to create fake news resistant societyPLOS ONE

Dear Dr. Tariq,

Thank you for submitting your manuscript to PLOS ONE. After careful consideration, we feel that it has merit but does not fully meet PLOS ONE’s publication criteria as it currently stands. Therefore, we invite you to submit a revised version of the manuscript that addresses the points raised during the review process. Please submit your revised manuscript by Aug 21 2024 11:59PM. If you will need more time than this to complete your revisions, please reply to this message or contact the journal office at plosone@plos.org. Please include the following items when submitting your revised manuscript:A rebuttal letter that responds to each point raised by the academic editor and reviewer(s). You should upload this letter as a separate file labeled 'Response to Reviewers'.A marked-up copy of your manuscript that highlights changes made to the original version. You should upload this as a separate file labeled 'Revised Manuscript with Track Changes'.An unmarked version of your revised paper without tracked changes. You should upload this as a separate file labeled 'Manuscript'.If applicable, we recommend that you deposit your laboratory protocols in protocols.io to enhance the reproducibility of your results. Protocols.io assigns your protocol its own identifier (DOI) so that it can be cited independently in the future. For instructions see: https://journals.plos.org/plosone/s/submission-guidelines#loc-laboratory-protocols. Additionally, PLOS ONE offers an option for publishing peer-reviewed Lab Protocol articles, which describe protocols hosted on protocols.io. Read more information on sharing protocols at https://plos.org/protocols?utm_medium=editorial-email&utm_source=authorletters&utm_campaign=protocols.

We look forward to receiving your revised manuscript.

Kind regards,

Fredrick Romanus Ishengoma

Academic Editor

PLOS ONE

Journal Requirements:

Reviewers' comments:

Reviewer's Responses to Questions

**Comments to the Author**

1. If the authors have adequately addressed your comments raised in a previous round of review and you feel that this manuscript is now acceptable for publication, you may indicate that here to bypass the “Comments to the Author” section, enter your conflict of interest statement in the “Confidential to Editor” section, and submit your "Accept" recommendation.

Reviewer #3: All comments have been addressed

Reviewer #5: All comments have been addressed

2. Is the manuscript technically sound, and do the data support the conclusions?

Reviewer #3: Partly

Reviewer #5: Yes

3. Has the statistical analysis been performed appropriately and rigorously? 

Reviewer #3: Yes

Reviewer #5: Yes

4. Have the authors made all data underlying the findings in their manuscript fully available?

Reviewer #3: Yes

Reviewer #5: Yes

5. Is the manuscript presented in an intelligible fashion and written in standard English?

Reviewer #3: Yes

Reviewer #5: Yes

6. Review Comments to the Author

Reviewer #3: Thanks for adding the potential usage of generative AI tools such as GPT models at the end of discussion. Again, when dealing with fake news detection, the state-of-the-art technique of deep learning might be constantly changing. The author(s) need to acknowledge how other scholars have used LLMs (e.g., BERT model, Roberta, longormer, etc.) in the field for fake news detection (which is absent in the current version).

Reviewer #5: - Abstract: Can the authors begin by briefly introducing the significant problem of fake news to emphasize the impact of this study since the beginning?

- Table 1: Table 1 will be very useful for future researchers. However, the authors should consider the consistency of the summarized data in the table. Some cells provided very detailed details, while others showed only the techniques' names.

- Data set: The retrieving date of the data should be mentioned.

- Data set: Some explanation of the data set should be added. For example, when and how Kaggle collected them.

- Since there is a keyword "Computational Thinking," this term should be used or mentioned in the contents.

7. PLOS authors have the option to publish the peer review history of their article (what does this mean?). If published, this will include your full peer review and any attached files.

Reviewer #3: No

Reviewer #5: No

---

## [Decision Letter · Decision Letter 3]

29 Oct 2024

PONE-D-23-40611R3Machine and deep learning algorithms for sentiment analysis during COVID-19: a vision to create fake news resistant societyPLOS ONE

Dear Dr. Tariq,

Thank you for submitting your manuscript to PLOS ONE. After careful consideration, we feel that it has merit but does not fully meet PLOS ONE’s publication criteria as it currently stands. Therefore, we invite you to submit a revised version of the manuscript that addresses the points raised during the review process.

Please highlight with color or do track changes for the revised parts in the manuscript in order to allow the reviewer to see the modification.

We look forward to receiving your revised manuscript.

Kind regards,

Fredrick Romanus Ishengoma

Academic Editor

PLOS ONE

Journal Requirements:

Reviewers' comments:

Reviewer's Responses to Questions

**Comments to the Author**

1. If the authors have adequately addressed your comments raised in a previous round of review and you feel that this manuscript is now acceptable for publication, you may indicate that here to bypass the “Comments to the Author” section, enter your conflict of interest statement in the “Confidential to Editor” section, and submit your "Accept" recommendation.

Reviewer #2: All comments have been addressed

Reviewer #5: All comments have been addressed

2. Is the manuscript technically sound, and do the data support the conclusions?

Reviewer #2: Yes

Reviewer #5: Yes

3. Has the statistical analysis been performed appropriately and rigorously? 

Reviewer #2: Yes

Reviewer #5: Yes

4. Have the authors made all data underlying the findings in their manuscript fully available?

Reviewer #2: Yes

Reviewer #5: Yes

5. Is the manuscript presented in an intelligible fashion and written in standard English?

Reviewer #2: Yes

Reviewer #5: Yes

6. Review Comments to the Author

Reviewer #2: All the changes suggested by the reviwers are positively addredesed by thr authors.

But, I will suggest authors to proof read the manuscript carefully.

For example: " 2.41. Support Vector Machines ". It will be 2.4.1. Similarrly

Proper abbrebration is required in manuscript. For example : iN section 2.4.1 abbrebration of SVM is given thrice.

Reviewer #5: Please highlight with color or do track changes for the revised parts in the manuscript in order to allow the reviewer to see the modification.

7. PLOS authors have the option to publish the peer review history of their article (what does this mean?). If published, this will include your full peer review and any attached files.

Reviewer #2: No

Reviewer #5: No

---

## [Author Response · Author response to Decision Letter 3]

13 Nov 2024

The attachment is within the submission system.

---

## [Decision Letter · Decision Letter 4]

26 Nov 2024

Machine and deep learning algorithms for sentiment analysis during COVID-19: a vision to create fake news resistant society

PONE-D-23-40611R4

Dear Dr. Tariq,

We’re pleased to inform you that your manuscript has been judged scientifically suitable for publication and will be formally accepted for publication once it meets all outstanding technical requirements.

Kind regards,

Fredrick Romanus Ishengoma

Academic Editor

PLOS ONE

Additional Editor Comments (optional):

Reviewers' comments:

Reviewer's Responses to Questions

**Comments to the Author**

1. If the authors have adequately addressed your comments raised in a previous round of review and you feel that this manuscript is now acceptable for publication, you may indicate that here to bypass the “Comments to the Author” section, enter your conflict of interest statement in the “Confidential to Editor” section, and submit your "Accept" recommendation.

Reviewer #5: All comments have been addressed

2. Is the manuscript technically sound, and do the data support the conclusions?

Reviewer #5: Yes

3. Has the statistical analysis been performed appropriately and rigorously? 

Reviewer #5: Yes

4. Have the authors made all data underlying the findings in their manuscript fully available?

Reviewer #5: Yes

5. Is the manuscript presented in an intelligible fashion and written in standard English?

Reviewer #5: Yes

6. Review Comments to the Author

Reviewer #5: Thank you for the response. The authors have responded to the comments. The reviewer considered this manuscript can be accepted.

7. PLOS authors have the option to publish the peer review history of their article (what does this mean?). If published, this will include your full peer review and any attached files.

Reviewer #5: No

---

## [Editor Report · Acceptance letter]

5 Dec 2024

PONE-D-23-40611R4 

PLOS ONE

Dear Dr. Tariq, 

I'm pleased to inform you that your manuscript has been deemed suitable for publication in PLOS ONE. Congratulations! Your manuscript is now being handed over to our production team.

Kind regards, 

on behalf of

Dr. Fredrick Romanus Ishengoma 

Academic Editor

PLOS ONE